# GLATrack: Global and Local Awareness for Open-Vocabulary Multiple Object Tracking

### Guangyao Li
Key Laboratory of Multimedia Trusted Perception and Efficient Computing, Ministry of Education of China, Xiamen University
Xiamen, China
Fujian Key Laboratory of Sensing and Computing for Smart City, School of Informatics, Xiamen University
Xiamen, China
liguangyao@stu.xmu.edu.cn

### Yajun Jian
Key Laboratory of Multimedia Trusted Perception and Efficient Computing, Ministry of Education of China, Xiamen University
Xiamen, China
Fujian Key Laboratory of Sensing and Computing for Smart City, School of Informatics, Xiamen University
Xiamen, China
yajunjian@stu.xmu.edu.cn

### Yan Yan
Key Laboratory of Multimedia Trusted Perception and Efficient Computing, Ministry of Education of China, Xiamen University
Xiamen, China
Fujian Key Laboratory of Sensing and Computing for Smart City, School of Informatics, Xiamen University
Xiamen, China
yanyan@xmu.edu.cn

### Hanzi Wang*
Key Laboratory of Multimedia Trusted Perception and Efficient Computing, Ministry of Education of China, Xiamen University
Xiamen, China
Fujian Key Laboratory of Sensing and Computing for Smart City, School of Informatics, Xiamen University
Xiamen, China
hanzi.wang@xmu.edu.cn

## ABSTRACT
Open-vocabulary multi-object tracking (MOT) aims to track arbitrary objects encountered in the real world beyond the training set. However, recent methods rely solely on instance-level detection and association of novel objects, which may not consider the valuable fine-grained semantic representations of the targets within key and reference frames. In this paper, we propose a Global and Local Awareness open-vocabulary MOT method (GLATrack), which learns to tackle the task of real-world MOT from both global and instance-level perspectives. Specifically, we introduce a region-aware feature enhancement module to refine global knowledge for complementing local target information, which enhances semantic representation and bridges the distribution gap between the image feature map and the pooled regional features. We propose a bidirectional semantic complementarity strategy to mitigate semantic misalignment arising from missing target information in key frames, which dynamically selects valuable information within reference frames to enrich object representation during the knowledge distillation process. Furthermore, we introduce an appearance richness measurement module to provide appropriate representations for targets with different appearances. The proposed method

gains an improvement of 6.9% in TETA and 5.6% in mAP on the large-scale TAO benchmark.

## CCS CONCEPTS
• **Computing methodologies → Tracking**.

## KEYWORDS
Open-vocabulary Multi-object Tracking; Region-aware Feature Enhancement; Appearance Richness Measurement; Bidirectional Semantic Complementarity

**ACM Reference Format:**
Guangyao Li, Yajun Jian, Yan Yan, and Hanzi Wang. 2024. GLATrack: Global and Local Awareness for Open-Vocabulary Multiple Object Tracking. In *Proceedings of the 32nd ACM International Conference on Multimedia (MM '24), October 28-November 1, 2024, Melbourne, VIC, AustraliaProceedings of the 32nd ACM International Conference on Multimedia (MM'24), October 28-November 1, 2024, Melbourne, Australia.* ACM, New York, NY, USA, 10 pages. https://doi.org/10.1145/3664647.3681530

## 1 INTRODUCTION
Multiple Object Tracking (MOT) is recognized as a fundamental task in computer vision, primarily aimed at detecting and tracking objects in a video sequence. This task plays a crucial role in various applications, such as autonomous driving [20, 39] and video analysis [29, 48]. While MOT methods [6, 15, 34, 54, 63] have made significant progress in recent years, they are typically trained on predefined data distributions. Additional annotations are required to track unseen targets, which not only restricts the scope of target perception but also hampers the ability to track novel objects in the visual world. Consequently, a disparity emerges between performance evaluation and real-world applications.

*Corresponding author: Hanzi Wang (hanzi.wang@xmu.edu.cn)

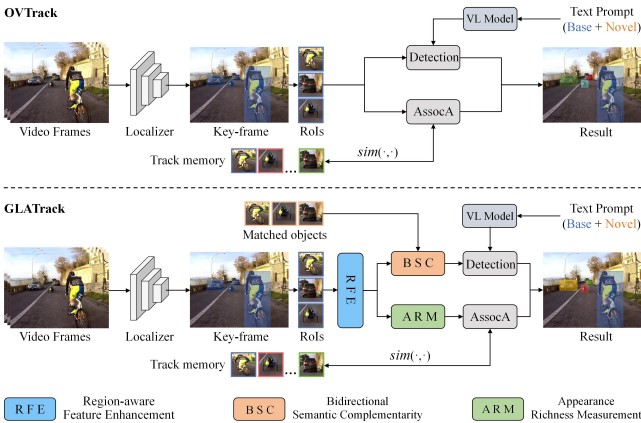

**Figure 1: Comparison of inference frameworks between OV-Track and GLATrack. Both methods track unseen objects by querying the discriminative vision-language model. The color of the bounding box indicates the track ID of the object.**

Several works [24, 30, 31] attempt to address such limitations by leveraging an open-world context. Ošep et al. [30] integrate a classifier following the tracking of detections to enhance general tracking capability in scenarios where no semantic information is accessible. Liu et al. [24] establish the concept of open-world tracking and perform both open and closed-world tracking. Recently, visual language models (VLMs) pre-trained on large-scale image-text pairs, such as CLIP [33], have shown remarkable classification capability in various visual tasks. Li et al. [23] first introduces CLIP as an open-vocabulary classifier to detect unseen categories and propose a data hallucination strategy employing static images to achieve data association. However, there are three inherent difficulties to this setup. First, using CLIP directly in MOT is suboptimal, as shown in Fig. 1. This is because the region features acquired by RoI Align [18] suffer from a significant loss of contextual information, exacerbating the fine-grained differences between image regions and text spans. Second, the semantic information in the appearance of targets exhibits diversity. Therefore, using the same dimension to represent objects with different appearance semantic richness is unreasonable. Third, the adjacent frames over a video contain coherent trajectories and semantic information about the targets. However, only the target's information from the key frames is utilized during the classification phase, neglecting the semantic information available in the adjacent frames. This underutilization hinders the potential of CLIP in the task of open-vocabulary MOT.

In this paper, we introduce a novel open-vocabulary MOT method, termed GLATrack, which aims to track unseen objects beyond the pre-defined training categories from both global and instance-level perspectives. Specifically, we devise a plug-and-play Region-aware Feature Enhancement (RFE) module that can be seamlessly integrated into the tracking pipeline, capturing global features from the FPN to enhance the region features. In detail, the RoI Align with the max pooling suffers from the loss of details, while Transformer [40] has considerable potential in capturing contextual information. To address this issue, we leverage a Transformer encoder to establish global relationships, with region features serving as the queries of

the decoder. This enables the region features to perceive and comprehend the semantic characteristics based on global features. As a result, we can obtain more discriminative semantic representations for the objects to strengthen the detection and tracking ability.

To further improve the capability of GLATrack in tracking generic unknown objects without annotation information, we introduce an Appearance Richness Measurement (ARM) module. This module facilitates fine-grained transformations from small to large dimensions, providing scalable representations for diverse target appearances based on their semantic richness and basic attributes. The ARM module is highly adaptable, avoiding assumptions about fixed representation dimension, allowing the model to acquire essential clues through self-learning for accurate judgment autonomously.

In addition to the localization and association branches, the classification pipeline also plays a crucial role in open-vocabulary MOT, which requires identifying targets in scenes with complex camera and object motion patterns. To effectively address these challenges, we propose a Bidirectional Semantic Complementarity (BSC) strategy to optimize semantic representations in key frames. Specifically, the reference frame contains the appearances of objects at different time points, which could provide valuable cues for classification. Consequently, we utilize CLIP to extract semantic information about the matching targets from the reference frame and employ it to filter out erroneous information in key frames that may interfere with object recognition. Subsequently, the refined representations are used to determine the categories of the targets.

The main contributions are summarized as follows:

• We propose a new open-vocabulary MOT method, GLATrack, which detects and tracks objects of arbitrary categories in the real world from both global and instance-level perspectives.

• We propose a flexible plug-and-play RFE module, which effectively leverages global information from the feature pyramid network to complement local targets by leveraging the context modeling ability of the Transformer.

• We propose a novel ARM module to generate adaptive appearance representations for objects with diverse semantic richness. Additionally, we introduce a BSC strategy, leveraging valuable information within reference frames to address classification biases arising from occluded object appearances in key frames.

• Experimental results demonstrate that the proposed method outperforms the state-of-the-art methods on the large-vocabulary MOT benchmark TAO, showing significant improvement on both TETA and mAP metrics.

## 2 RELATED WORK

**Multi-object Tracking.** Some MOT methods [5, 32, 37, 38, 41, 44, 51] typically adopt the tracking-by-detection paradigm [1], which involve two main stages: detection and association. SORT [4] utilizes Kalman filters and the Hungarian algorithm for motion prediction and frame-to-frame data association. ByteTrack [60] leverages the correlation between detection boxes and tracking trajectories to maintain high-confidence detection results and restore low-confidence detections. Recently, numerous studies have concentrated on utilizing Transformer to improve motion estimation. For example, MOTR [57] and Trackformer [27] introduce the Transformer to integrate detection and tracking into a query-based

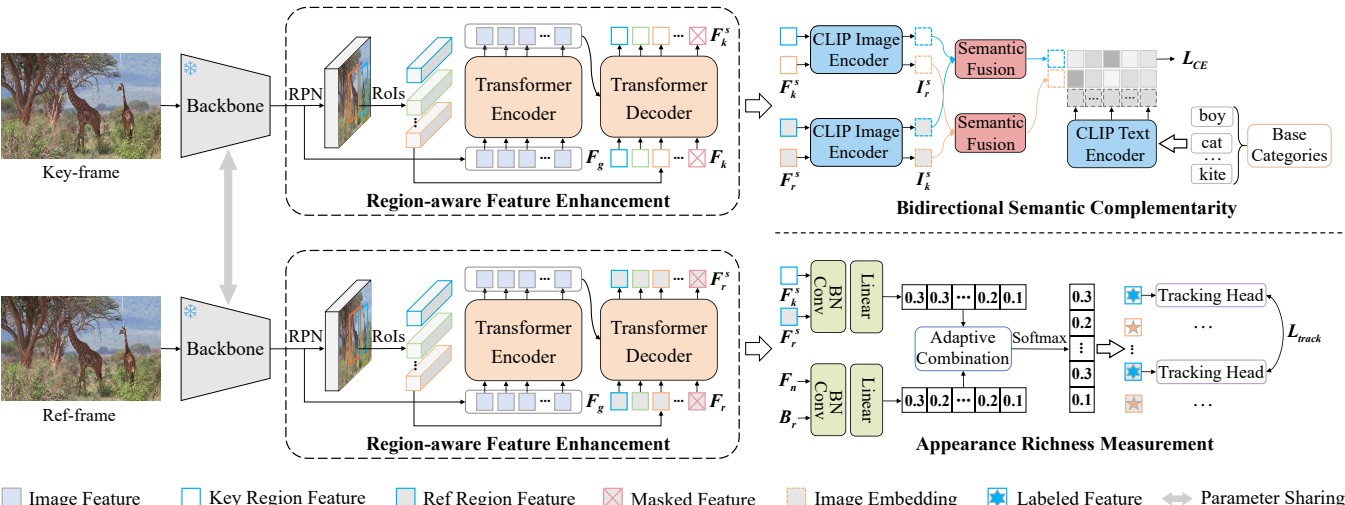

**Figure 2: The architecture of the proposed GLATrack. Initially, given the key and reference frames, RoIs are extracted from the backbone through an RPN. These RoIs are then fed into the RFE module for feature enhancement. Subsequently, the enhanced RoIs are sent to the BSC module, where the image embeddings within the key frames are complemented with those within the reference frame before undergoing the classification process. To facilitate data association, the enhanced RoIs are represented at various scales based on their semantic information scores, which are calculated by the proposed ARM module.**

framework. Most methods mentioned above work well at detecting and tracking a closed set of categories present in training datasets, while performing blindly in unseen categories in inference. To address this limitation, Dave et al. [10] introduce a large-scale and diverse benchmark TAO [10]. Based on this benchmark, GTR [68] takes a brief sequence of frames as input and computes the global trajectories for all objects. TET [22] computes and compares class exemplars for each localized object to identify potential matching candidates, then conducts class-agnostic association based solely on instance appearance features. However, these methods still suffer from limitations due to pre-defined object categories. Instead, the proposed method aims to achieve effective tracking of novel objects within an open-vocabulary context.

**Open-Vocabulary Detection and Tracking.** Open-vocabulary learning is widely applied in the field of computer vision [21, 42, 49, 50, 53, 58, 65, 66], which enables the recognition of unseen classes by large vocabulary knowledge. OVR-CNN [56] proposes the concept of open-vocabulary object detection, where image captions are utilized to pre-train the text encoder. Recently, based on the TAO dataset [10], TAO-OW [24] is proposed as a new benchmark for open-world tracking. OWL-ViT [19] utilizes the Transformer decoder to propagate representations through time, enabling the open-vocabulary detection model [28] to adapt to videos. To transcend pre-defined categories and achieve open-vocabulary MOT, OVTrack [23] leverages CLIP to locate and classify unseen targets. However, that method still suffers from the limitation that using static image classification neglects valuable semantic information in a given video sequence. In contrast, the proposed GLATrack tries to leverage the semantic information contained in adjacent frames to improve the accuracy of object classification.

**Learning Tracking from Static Images.** Recently, several methods [14, 46, 59, 61, 67] employ static images directly to train the

track head in MOT and achieve promising performance. For example, KDMOT [59] utilizes a pre-trained teacher model to distill general recognition capabilities into the Re-ID head during the training while maintaining an efficient architecture during inference. QDTrack [14] proposes a quasi-dense similarity learning strategy, achieving competitive tracking performance at the static image level through contrastive learning for the association process. Li et al. [23] introduce a data hallucination strategy based on denoising diffusion probabilistic models [36] to generate pseudo-LVIS videos and apply a fixed Re-ID [62, 64] feature dimension for similarity learning. However, it is less reasonable as the semantic richness of the objects in LVIS that contains 1203 categories varies greatly. Therefore, we propose an adaptive encoding network that encodes the objects with different appearance richness into corresponding Re-ID features for data association.

**Knowledge Distillation.** Knowledge distillation techniques aim to distill the knowledge of Vision-and-Language Models (VLMs) into specific tasks [2, 7, 13, 26]. ViLD [17] employs instance-level visual-to-visual knowledge distillation to integrate open-vocabulary knowledge into a two-stage detector. HierKD [26] introduces a global-level knowledge distillation module, which aligns global-level image representations with caption semantics. OADP [43] employs instance-level distillation along with global and block distillation methods to construct relationship information among objects during distillation. PCL [8] adopts an image captioning model to generate detailed labels that describe object instances from multiple perspectives. Instead of directly applying image knowledge distillation to open-vocabulary MOT, our GLATrack exploits the auxiliary role of the reference frames to aggregate the features of the same target across various spatiotemporal contexts, thereby obtaining precise and comprehensive object information.

# 3 METHODOLOGY

## 3.1 Overview

The overview architecture of the proposed method is shown in Fig. 2. GLATrack is designed to track objects beyond predefined categories. Initially, global features $F_g$ are extracted by a pre-trained backbone, and then the RPN generates region proposals, which are fed into the RoI Align to produce fixed-size features denoted as $F_r$. Based on $F_g$, the RFE module enables the region features $F_r$ to retrieve and restore their prominent characteristics that may have been lost during the pooling process. Utilizing the enhanced features $F^s$ as input, the ARM module and BSC strategy are employed to improve tracking and classification performance, respectively.

## 3.2 Problem Formulation

In the training phase, video sequences $\mathbf{X}^{\text{train}}$ and the corresponding annotations $\mathbf{A}^{\text{train}}$ including base object categories $C^{\text{base}} \subset \mathbb{N}$ are provided. The input $x_i \in \mathbf{X}^{\text{train}}$ is fed into the backbone with RoI Align to obtain the region features $F_r$, which are then processed by the RFE module to obtain refined features $F^s$. During classification, the distilled CLIP image encoder $\mathcal{E}_d$ takes the $F^s$ as input to generate image embeddings $\mathcal{I}$, which are then compared with text embeddings $\mathcal{A}$ using cosine similarity to determine the category of the $F^s$. For data association, $F^s$ is encoded by the ARM module into image embeddings with different dimensions and sent to the tracking head for training through contrastive learning.

During the inference phase, the open-vocabulary tracker aims to track the objects in the given video sequences $\mathbf{X}^{\text{test}}$ belonging to the classes $C^{\text{base}} \cup C^{\text{novel}}$. Each trajectory $\tau_t$ includes the state $\{id, \hat{l}_t, \hat{p}_t, \hat{c}_t\}$, where $\hat{p}_t$ represents the predicted object confidence score, $id$ indicates the track id, and $\hat{c}_t$ denotes the object class.

## 3.3 Region-aware Feature Enhancement

The input of the RFE module is the region proposals generated by RPN for each frame in the video sequences. Specifically, given the key frame $\mathbf{t}$, RPN generates the object candidates $\mathcal{R}(n) = \{\mathbf{r}_1, \mathbf{r}_2, ..., \mathbf{r}_n\}$, represented by the 2D bounding box coordinate $\mathbf{b} = [x, y, w, h]$ and the corresponding object confidence score $\mathbf{p} \in [0, 1]$. Aligning the variable number of detected targets in each frame to the same dimensions becomes challenging. To address this, we introduce a learnable query $\mathcal{M}$ of size $N \times D$, $N$ is the max number of objects in all batches and $D$ equals $w \times h$.

$$\mathcal{M} = [\mathbf{q}_1, ..., \mathbf{q}_n, \mathbf{m}_1, ..., \mathbf{m}_{N-n}], \tag{1}$$

where $n$ indicates the number of objects within a batch, $\mathbf{q}_i$ denotes the region query corresponding to $\mathbf{r}_i$, $\mathbf{m}_j$ is the masked query. For the global feature $F_g$, we employ a Transformer encoder to model the global relationship further, obtaining refined features denoted as $F_e$, which are used to enhance the region features. Technically, we extend region candidates $\mathcal{R}(n)$ to region queries within the Transformer, and the representations of these queries are updated in the Transformer decoder. Specifically, region queries are updated through cross-attention with $F_e$.

$$F^s = Softmax(\frac{\mathcal{M}F_e^{\top}}{\sqrt{d_e}})F_e, \tag{2}$$

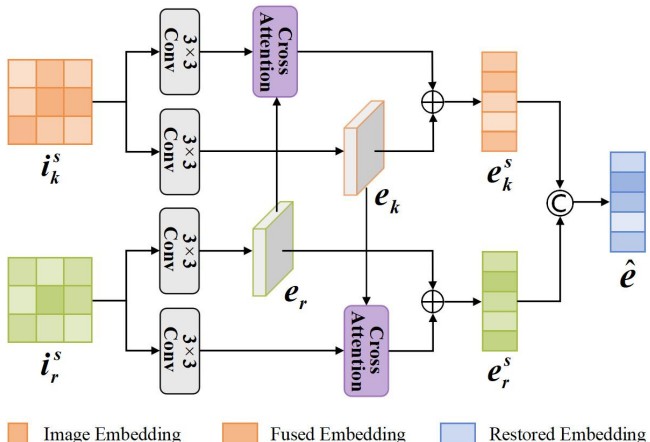

Figure 3: The architecture of the BSC module. It achieves efficient image embedding selected from the key and reference frames through a stack of convolutions and cross-attention.

where the masked queries generate values during the update process. These masked queries will be re-masked after each decoder layer to prevent them from influencing region queries. The updated region queries are further translated to enhance region features $\mathbf{f}_i^s$. Such a way ensures that region features capture the global image context comprehensively, significantly reducing the disparity between the entire image and region distributions.

## 3.4 Bidirectional Semantic Complementarity

The existing open-vocabulary MOT method, OVTrack [23], connects the Faster R-CNN with CLIP directly, which ignores the rich target information that adjacent frames may contain. To alleviate this limitation, the BSC strategy is proposed to incorporate feature information from the reference frame into the CLIP classification process, as shown in Fig. 3.

To accelerate inference speed, we employ the CLIP image encoder to supervise the training of the image head, which generates the embeddings $\mathcal{I} = \{\mathbf{i}_k^s, \mathbf{i}_r^s\} \in \mathbb{R}^{H \times W \times d}$ for each enhanced feature $f^s$. To ensure the image embedding fusion at a more granular level, we separately apply 3×3 filters to $\mathbf{i}_k$ and $\mathbf{i}_r$ to derive deeper-level embeddings $\mathbf{e}_k$ and $\mathbf{e}_r$. Subsequently, three independent full-connected layers transform the visual features as $Q$, $K$, and $V$:

$$Q = W_q(e_k + p_k) \in \mathbb{R}^{HW \times d},$$
$$K = W_k(e_r + p_r) \in \mathbb{R}^{HW \times d}, \tag{3}$$
$$V = W_v(e_r + p_r) \in \mathbb{R}^{HW \times d},$$

where $W_q, W_k, W_v$ are the learnable weight, $p_k$ and $p_r$ are the position embeddings, respectively. We then perform matrix multiplication with $Q$, $K$, and $V$, followed by weighting the language features using the resulting similarity matrix. Subsequently, the $e_k$ are combined with the key frame features influenced by the reference frame to obtain the fused embeddings $\mathbf{e}_k^s \in \mathbb{R}^{HW \times d}$:

$$\mathbf{e}_k^s = Softmax(\frac{QK^{\top}}{\sqrt{d}})V + \mathbf{e}_k, \tag{4}$$

where $d$ is the feature dimension. Similarly, the features of the reference frame undergo the process mentioned above to produce the $\mathbf{e}_r^s \in \mathbb{R}^{HW \times d}$.

$$\hat{\mathbf{e}} = Decoder(concat(\mathbf{e}_k^s, \mathbf{e}_r^s)). \tag{5}$$

By concatenating $\mathbf{e}_k^s$ and $\mathbf{e}_r^s$ along the channel dimension and decoding them, restored embeddings $\hat{\mathbf{e}} \in \mathbb{R}^{HW \times d}$ are aligned with the counterpart embeddings generated by the CLIP image encoder.

## 3.5 Appearance Richness Measurement

Current MOT methods in open-vocabulary environments heavily rely on appearance cues for data association where motion cues are delicate due to various camera and object motion patterns. However, restricting target appearance to uniform dimensions hampers the capability of the model to grasp the intrinsic semantic information in these appearances, making learning robust target appearances challenging. To address this issue, the proposed ARM module characterizes the target appearance using specific feature dimensions corresponding to their appearance semantic richness. Specifically, given the input image pairs $(x_{key}, x_{ref})$, we extract corresponding enhanced RoIs $(F_k^s, F_r^s)$. As each $f^s \in F^s$ encapsulates semantic features of the target, we convert it into a semantic assessment score $score_t$.

$$score_t = Linear(Reshape(Conv(f^s))) \in \mathbb{R}^{1 \times D}, \tag{6}$$

where $D$ is the predefined maximum representation scale. However, as different $f^s$ are represented with equal information content, a single $score_t$ cannot fully capture its informational significance in the input image. Therefore, we introduce $scores_o$, derived from the fundamental information $N(w, h, u, r)$ of $f^s$ in the original image, where $w$ and $h$ represent the width and height, $u$ indicates the area and $r$ denotes the proportion to the entire image. Subsequently, $score_t$ is adaptively combined with the computed $scores_o$.

$$score_d = \lambda score_t + (1 - \lambda)score_o. \tag{7}$$

where $\lambda$ serves as a trainable parameter. Finally, $score_d$ is utilized to determine the representation scale, with the position associated with the highest score indicating the scale of $f^s$.

## 3.6 Training and Inference

During the process of transferring CLIP knowledge to our task, we utilize L1 loss to supervise the training of the image head. The goal is to minimize the distance between $\hat{\mathbf{e}}_i$ and $\beta_i$.

$$\mathcal{L}_{\text{image}} = \frac{1}{|\mathcal{R}|} \sum_{r \in \mathcal{R}} ||\hat{\mathbf{e}}_i - \beta_i||_1, \tag{8}$$

where $\beta_i$ is the corresponding image embedding encoded by the CLIP image encoder. For the class label of $f^s$, we generate text prompts $\mathcal{A}(\hat{\mathbf{c}})$, which consists of $L$ context vectors $\mathbf{c}^v$ and a class name embedding $\mathbf{c}^n$. Then the CLIP text encoder $\mathcal{E}$ takes the input of the prompt and outputs the corresponding text embedding $\mathbf{t}_c$, $\forall \mathbf{c} \in C^{\text{base}}$. We calculate the similarity between the predicted embeddings $\mathbf{t}_p$ and their corresponding CLIP embeddings $\mathbf{t}_c$ to synchronize them.

$$\mathbf{V}(r) = Softmax[\cos(\mathbf{t}_p, \mathbf{t}_b)/\lambda, \cos(\mathbf{t}_p, \mathbf{t}_{|C^{\text{base}}|})/\lambda], \tag{9}$$

where $cos(\cdot, \cdot)$ indicates the cosine similarity, $\lambda$ denotes a temperature parameter and $\mathbf{t}_b$ represents a learned background prompt.

$$\mathcal{L}_{\text{text}} = \frac{1}{|\mathcal{R}|} \sum_{r \in \mathcal{R}} \mathcal{L}_{\text{CE}}(\mathbf{V}(r), c_r), \tag{10}$$

The cross-entropy loss $\mathcal{L}_{\text{CE}}$ is calculated using $cr$ as the class label of $r$. We employ contrast learning to achieve robust tracking based on appearance cues. For the matched RoIs in the reference frame $x_{ref}$, we group objects with the same identity into $K^+$ and separate objects with different identities into $K^-$.

$$\text{Sim}(\mathbf{r}) = \frac{\exp(r \cdot \mathbf{k}^+/\tau)}{\alpha \sum_{K^+} \exp(r \cdot \mathbf{k}^+/\tau) + \sum_{K^-} \exp(r \cdot \mathbf{k}^-/\tau)}, \tag{11}$$

$$\mathcal{L}_{\text{track}} = -\sum_{r \in \mathcal{R}} \frac{1}{|K^+(r)|} \sum_{\mathbf{k}^+ \in K^+(r)} \log(\text{Sim}(\mathbf{k}^+)), \tag{12}$$

where $\alpha$ equals $\frac{1}{|K^+(\mathbf{k})|}$. $k$ is feature embedding of the training sample in the key frame, $k^+$ and $k^-$ denote its positive and negative targets, respectively. During the inference phase, given the enhanced RoIs, existing trajectories $\mathcal{T}$ are directly associated with targets $r \in \mathcal{R}$ using appearance feature similarity. The similarity between targets $\tau$ within trajectories and candidate targets $r$ is measured using both bidirectional softmax and cosine similarity.

$$\mathbf{f}(\tau, r) = \frac{1}{2} \left[ \frac{\exp(\mathbf{u}_r \cdot \mathbf{v}_\tau)}{\sum_{r' \in \mathcal{R}} \exp(\mathbf{u}_{r'} \cdot \mathbf{v}_\tau)} + \frac{\exp(\mathbf{u}_r \cdot \mathbf{v}_\tau)}{\sum_{\tau' \in \mathcal{T}} \exp(\mathbf{u}_r \cdot \mathbf{v}_{\tau'})} \right]. \tag{13}$$

The higher bi-softmax value indicates that the distance between the two matched objects in the feature space is reduced. If $\mathbf{f}(\tau, r) > \alpha$, the target with the maximum similarity is added to the current trajectory. If objects $r$ fail to match any existing tracks but maintain a confidence score higher than $\sigma$, a new track will be initiated. In the classification pipeline, the enhanced features $f^s$ serve as inputs to the distilled image head, generating corresponding image embeddings $\hat{\mathbf{e}}$. These image embeddings, along with generated text embeddings, are used to compute the similarity for determining the category of the targets.

## 4 EXPERIMENTS

### 4.1 Datasets

We conduct experiments on the publicly available large-vocabulary MOT benchmark TAO [10]. GLATrack is trained on the hallucinated LVIS dataset [23] within the base categories. We evaluate GLATrack on both the validation and test sets of TAO.

**LVIS Dataset.** Unlike MOT datasets [9, 11, 55] with limited object classes, LVIS offers annotations for a large vocabulary of object categories, covering a wide spectrum of real-world objects. Following the OVTrack [23], we use their hallucinated LVIS dataset. Compared with TAO, this dataset simulates object motion in an open-world context with a smaller dataset size, making it suitable for open-vocabulary MOT tasks, especially in scenarios with limited computational resources. In this dataset, we employ the 886 frequent and common categories as the base classes $C^{base}$, while reserving the 337 rare categories as the novel categories $C^{novel}$.

**TAO Dataset.** The TAO dataset is a comprehensive video dataset designed to provide rich visual and semantic data for diverse video analysis tasks. A notable feature of the TAO dataset is its openness, allowing tracking of any object rather than being limited to specific

**Table 1: Comparison with state-of-the-art methods on both base and novel classes in the validation and test sets of TAO. All methods use ResNet-50 as the backbone. We indicate the classes and datasets of the training data for each method. Note that GLATrack and OVTrack are exclusively trained on the static LVIS dataset. The best results are highlighted in bold.**

| Method | Classes | | Data | | Base | | | | Novel | | | |
|---|---|---|---|---|---|---|---|---|---|---|---|---|
| Validation set | Base | Novel | LVIS | TAO | TETA | LocA | AssocA | ClsA | TETA | LocA | AssocA | ClsA |
| QDTrack [14] | ✓ | ✓ | ✓ | ✓ | 27.1 | 45.6 | 24.7 | 11.0 | 22.5 | 42.7 | 24.4 | 0.4 |
| TETer [22] | ✓ | ✓ | ✓ | ✓ | 30.3 | 47.4 | 31.6 | 12.1 | 25.7 | 45.9 | 31.1 | 0.2 |
| DeepSORT (ViLD) [45] | ✓ | − | ✓ | ✓ | 26.9 | 47.1 | 15.8 | 17.7 | 21.1 | 46.4 | 14.7 | 2.3 |
| Tracktor++ (ViLD) [3] | ✓ | − | ✓ | ✓ | 28.3 | 47.4 | 20.5 | 17.0 | 22.7 | 46.7 | 19.3 | 2.2 |
| OVTrack [23] | ✓ | − | ✓ | − | 35.5 | 49.3 | 36.9 | 20.2 | 27.8 | 48.8 | 33.6 | 1.5 |
| GLATrack (Ours) | ✓ | − | ✓ | − | **37.9** | **54.3** | **38.7** | **20.8** | **31.0** | **53.2** | **37.2** | **2.6** |
| Method | Classes | | Data | | Base | | | | Novel | | | |
| Test set | Base | Novel | LVIS | TAO | TETA | LocA | AssocA | ClsA | TETA | LocA | AssocA | ClsA |
| QDTrack [14] | ✓ | ✓ | ✓ | ✓ | 25.8 | 43.2 | 23.5 | 10.6 | 20.2 | 39.7 | 20.9 | 0.2 |
| TETer [22] | ✓ | ✓ | ✓ | ✓ | 29.2 | 44.0 | 30.4 | 10.7 | 21.7 | 39.1 | 25.9 | 0.0 |
| DeepSORT (ViLD) [45] | ✓ | − | ✓ | ✓ | 24.5 | 43.8 | 14.6 | 15.2 | 17.2 | 38.4 | 11.6 | 1.7 |
| Tracktor++ (ViLD) [3] | ✓ | − | ✓ | ✓ | 26.0 | 44.1 | 19.0 | 14.8 | 18.0 | 39.0 | 13.4 | 1.7 |
| OVTrack [23] | ✓ | − | ✓ | − | 32.6 | 45.6 | 35.4 | 16.9 | 24.1 | 41.8 | 28.7 | 1.8 |
| GLATrack (Ours) | ✓ | − | ✓ | − | **35.7** | **51.3** | **37.3** | **18.4** | **26.3** | **46.5** | **29.0** | **3.3** |

object categories. It consists of three partitions: train, validation, and test, encompassing 500, 988, and 1419 videos, respectively. In the test set of the TAO, 324 object classes overlap with the LVIS, and we define 33 of these classes as rare, which is denoted as $C^{novel}$.

## 4.2 Evaluation Metrics

**TETA.** TETA [22] is built upon the HOTA [25], extending its functionality to address challenges posed by long-tail scenarios. TETA is composed of three parts: localization score (LocA), classification score (ClsA), and association score (AssocA). The TETA score is obtained by the arithmetic mean of these three scores.

**Track mAP.** Track mAP [52] offers a comprehensive evaluation by considering both the detection accuracy and the ability to associate targets. This assessment involves measuring the overlap between the predicted bounding boxes and the ground truth.

## 4.3 Implementation Details

We use Faster R-CNN [35] with ResNet-50 [16] as the open-vocabulary detector. The model is trained on 4 GPUs, with 2 images per GPU. Our training process consists of two stages. Initially, we train the open-vocabulary object detector following [13, 17]. Then we fine-tune the model for tracking, assigning loss weights of 0.25 for $\mathcal{L}_{track}$ and 1.0 for $\mathcal{L}_{aux}$ based on [14, 23]. During training, non-maximum suppression (NMS) is applied to filter boxes $R$ with an IoU threshold of 0.7, and we randomly select $|R|$ boxes per image ($|R| = 256$). For inference, boxes with an IoU threshold of 0.5 are selected. We employ the SGD optimizer with an initial learning rate of 0.02 and set the weight decay and momentum to 0.0001 and 0.9, respectively. In terms of track management, each track maintains a history of 10 frames, with parameters $\alpha = 0.5$ and $\sigma = 0.0001$.

**Table 2: Comparison with Closed-set MOT methods in TETA, LocA, AssocA, and ClsA. We compare GLATrack with existing trackers on the TAO validation set. Both GLATrack and OVTrack employ ResNet-50 as the backbone, while other competing methods mainly utilize ResNet-101, except for AOA. All methods employ Faster R-CNN as the detector.**

| Method | TETA | LocA | AssocA | ClsA |
|---|---|---|---|---|
| SORT-TAO [10] | 24.8 | 48.1 | 14.3 | 12.1 |
| Tracktor [3] | 24.2 | 47.4 | 13.0 | 12.1 |
| DeepSORT [45] | 26.0 | 48.4 | 17.5 | 12.1 |
| AOA [12] | 25.3 | 23.4 | 30.6 | **21.9** |
| Tracktor++ [10] | 28.0 | 49.0 | 22.8 | 12.1 |
| QDTrack [14] | 30.0 | 50.5 | 27.4 | 12.1 |
| TETer [22] | 33.3 | 51.6 | 35.0 | 13.2 |
| OVTrack [23] | 34.7 | 49.3 | 36.7 | 18.1 |
| GLATrack | **37.1** | **54.1** | **38.5** | 18.6 |

## 4.4 Comparison with State-of-the-Art Methods

**Open-vocabulary MOT.** We compare GLATrack with other methods on the challenging MOT benchmark TAO [10]. Closed-set trackers, such as QDTrack [14] and TETer [22], are trained on the entire LVIS dataset, while DeepSORT [45] and Trackor++ [10] are integrated with the open-vocabulary detector ViLD [17] for open-vocabulary MOT. OVTrack [23], along with the aforementioned methods, are trained only on the base categories of LVIS. The evaluation results on the validation and test sets of TAO are presented in Tab. 1, which are divided into two parts: base categories and novel categories. The proposed method achieves significant performance

**Table 3: Comparison with Closed-set MOT methods in Track mAP50, Track mAP75, and Track mAP. We compare GLATrack with existing trackers on the validation set of TAO. ∗ indicates the use of static images for training.**

| Method | Track mAP50 | Track mAP75 | Track mAP |
|---|---|---|---|
| SORT-TAO [10] | 13.2 | - | - |
| QDTrack [14] | 15.9 | 5.0 | 10.6 |
| GTR* [68] | 20.4 | - | - |
| TAC [47] | 17.7 | 5.80 | 7.30 |
| BIV [46] | 19.6 | 7.30 | 13.6 |
| OVTrack* [23] | 21.2 | 10.6 | 15.9 |
| GLATrack* | **22.2** | **11.5** | **16.8** |

**Table 4: Ablation studies on the validation set of TAO to evaluate the contribution of modules in the proposed GLATrack.**

| Base | RFE | ARM | BSC | TETA | LocA | AssocA | ClsA |
|---|---|---|---|---|---|---|---|
| ✓ | | | | 34.7 | 49.3 | 36.7 | 18.1 |
| ✓ | ✓ | | | 36.2 | 54.1 | 37.2 | 17.2 |
| ✓ | | ✓ | | 34.9 | 49.5 | 37.3 | 17.6 |
| ✓ | ✓ | ✓ | | 36.3 | 53.8 | 37.9 | 17.3 |
| ✓ | ✓ | ✓ | ✓ | **37.1** | **54.1** | **38.5** | **18.6** |

on the benchmark, outperforming comparison methods across the base and novel categories on all metrics. Specifically, GLATrack achieves the highest score of 37.9 TETA in base classes and 31.0 TETA in novel classes on the validation set. Although QDTrack [14] and TETer [22] are trained on both $C^{base}$ and $C^{novel}$, their performance is unsatisfactory compared with GLATrack, which is trained only on base classes yet achieves superior detection results on both base and novel classes. Additionally, GLATrack outperforms the first open-vocabulary tracker, OVTrack [23], by a substantial margin, achieving +2.4 TETA on base classes and +3.2 TETA on novel classes, respectively. Similarly, GLATrack also achieves excellent performance on the TAO test set. The competitive results demonstrate that GLATrack can effectively tackle object tracking in complex scenarios.

**Closed-set MOT.** We evaluate GLATrack against existing methods on the TAO validation set using TETA and Track mAP, as shown in Tab. 2 and 3. GLATrack and OVTrack [23] are both trained on the base categories using static images, while other closed-set trackers are trained on video datasets, which include these unseen categories. As AOA [12] integrates multiple few-shot detection and Re-ID models trained on other datasets, it achieves a higher classification score than other methods. However, GLATrack outperforms previous methods in almost all evaluation metrics. Specifically, GLATrack achieves the highest TETA score of 37.1 and AssocA score of 38.5, surpassing OVTrack by 2.4 points in TETA and 1.8 points in AssocA. For Track mAP, GLATrack achieves 22.2 Track mAP50 and 11.5 Track mAP75 on the validation set, which are 1.0 points and 0.9 points higher than the baseline, respectively, resulting in an overall performance improvement of 0.9 points. These improvements show that GLATrack exhibits excellent performance

**Table 5: Zero-shot generalization. GLATrack is tested on the BDD100K [55] MOT validation split. We provide an explanation of the training data corresponding to each model. ∗ denotes logit masking of classes not present in BDD100K.**

| Method | Training | TETA | LocA | AssocA | ClsA |
|---|---|---|---|---|---|
| QDtrack* [14] | LVIS, TAO | 35.6 | 38.1 | 28.5 | 40.2 |
| TeTer* [22] | LVIS, TAO | 36.1 | 36.4 | 31.9 | 40.2 |
| QDTrack [14] | LVIS, TAO | 32.0 | 25.9 | 27.8 | 42.4 |
| TETer [22] | LVIS, TAO | 33.2 | 24.5 | 31.8 | 43.4 |
| OVTrack [23] | LVIS | 42.5 | 41.0 | 36.7 | 49.7 |
| GLATrack | LVIS | **45.2** | **42.8** | **40.6** | **52.4** |

compared with other methods, especially when trained solely on static images.

**Zero-shot Generalization.** We evaluate the ability of GLATrack to generalize zero-shot learning compared with closed-set trackers by conducting experiments on the MOT benchmark BDD100K [55]. In our experiments, we condition the proposed tracker on textual prompts containing class names sourced from the BDD100K dataset. Additionally, we report the results with masked logits for classes absent in BDD100K. Tab. 5 presents TETA results on the BDD100K dataset, where GLATrack achieves the best performance across all metrics. Specifically, compared with OVTrack, GLATrack outperforms 1.8 points in localization, 3.9 points in association, and 2.7 points in classification, respectively. These results demonstrate the robust performance of GLATrack, even when applied to a challenging large-scale benchmark.

## 4.5 Ablation Studies

We conduct ablation studies on the validation set of TAO to assess the effectiveness of the modules in the proposed GLATrack, using TETA as the main evaluation metrics for the ablation experiments.

**Impact of Region-aware Feature Enhancement Module.** We validate the effectiveness of the RFE module, as shown in Tab. 4. Compared with the baseline method, RFE improves TETA by 1.5 points, LocA by 4.8 points, and AssocA by 0.5 points. These significant improvements in metrics indicate that the proposed RFE module effectively improves the embeddings extracted by RoI Align from global image features. As illustrated in Fig. 5, GLATrack focuses on the bird and its environment. However, incorporating enhanced features into the frozen image head decreases ClsA by 0.9 points due to feature inconsistency.

**Impact of Appearance Richness Measurement Module.** The ARM module quantifies semantic information in diverse visual representations, as shown in Tab. 4. Compared with encoding the objects with uniform Re-ID feature dimensions, the ARM module results in a notable improvement of 0.7 in AssocA. The significant improvement in AssocA indicates the effectiveness of the proposed ARM module. Furthermore, we conduct extensive experiments to explore the impact of different representation scales on tracking performance. Fig. 6 shows that increasing representation dimensions initially improves basic category metrics but later decreases, while novel category metrics show the opposite trend. Given the long-tail distribution of the TAO dataset, excessively large or small

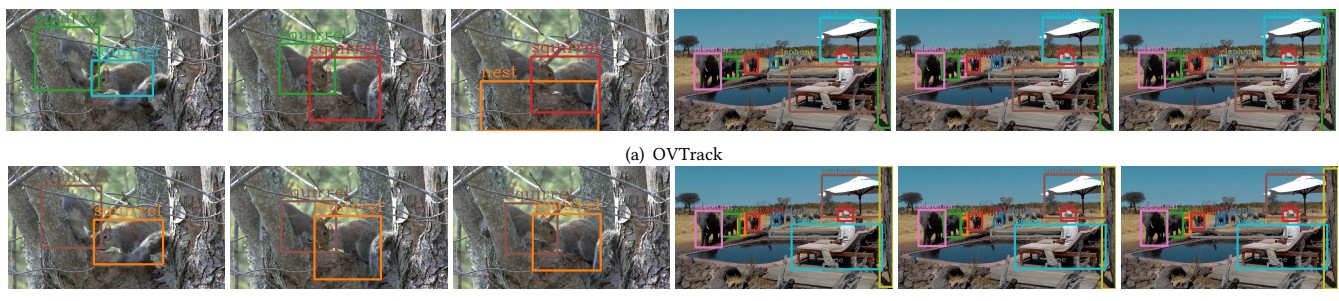

(a) OVTrack

(b) GLATrack

**Figure 4: Qualitative comparison between OVTrack [23] (top) and GLATrack (bottom) on the TAO dataset. Tracking targets from classes that are not encountered during the training. The identical box color indicates the same object.**

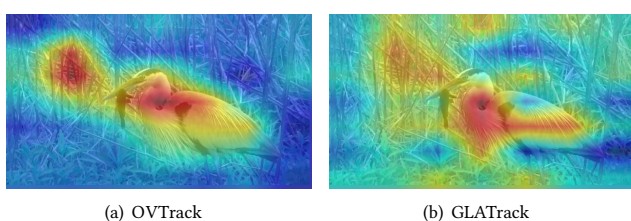

(a) OVTrack                           (b) GLATrack

**Figure 5: Two different heat maps obtained by OVTrack and GLATrack. OVTrack emphasizes the background, while GLATrack primarily focuses on the entire target and comprehends the surrounding environment.**

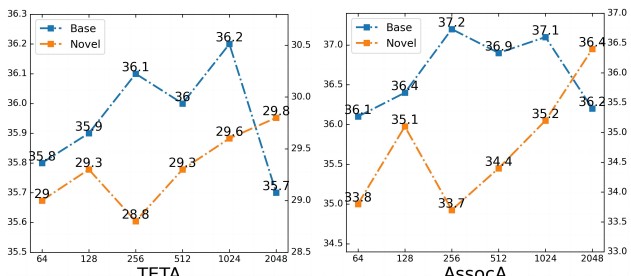

**Figure 6: The visualization of TETA and AssocA on the TAO dataset using different dimensions to represent object appearance, which contains base classes and novel classes.**

dimensions are not conducive to overall appearance representation. Therefore, we set the maximum dimension to 1024.

**Impact of Bidirectional Semantic Complementarity Strategy.** Replacing the original image head with BSC improves TETA by 0.8 points, LocA by 0.3 points, and ClsA by 1.5 points, as shown in the fourth row of Tab. 4. This indicates that supplementing semantic information further enhances classification accuracy. Specifically, compared to the original image head, BSC enhances the semantic representation of matching targets in key frames, which is beneficial to the model in identifying targets in complex scenes. Furthermore, the utilization of BSC leads to a notable enhancement in the classification metric involving both the first and second modules. This also explains the observed decrease in classification performance in ablation experiments.

### 4.6 Qualitative Analysis

The qualitative comparison results illustrated in Fig. 4 demonstrate the capabilities of both OVTrack and GLATrack in tracking novel targets. However, in the left side results, OVTrack encounters an ID switch and target loss when the squirrels are occluded and blend with the background color in the second and third frames, respectively. In contrast, GLATrack maintains a correct ID and successfully locates the targets. Additionally, the right side results depict challenges of fine-grained classification in target occlusion scenarios. OVTrack exhibits an unclear position of the elephant in the first frame and loses its position in the third frame, whereas GLATrack accurately localizes the target. We present the failure cases of the proposed method: throughout the tracking process,

GLATrack struggles to differentiate the tree trunk next to the pool due to limited valuable information, yet it maintains stable tracking.

## 5 CONCLUSION

In this paper, we propose GLATrack, a novel open-vocabulary multi-object tracking framework. Recognizing the importance of robust semantic representation, we designed an RFE module that utilizes the powerful contextual modeling capability of Transformer to enhance pooled region features. To efficiently and accurately detect novel objects, we propose a BSC module to distill knowledge from large-scale vision-language models, leveraging information from both key and reference frames. Additionally, we introduce an ARM module to determine the representation scale of the target based on fundamental attributes and semantic richness, facilitating robust data association relying on appearance cues. Extensive experiments on the large-scale, large-vocabulary TAO benchmark demonstrate that the proposed method outperforms existing trackers in detecting and tracking objects across both base and unseen novel classes, achieving state-of-the-art performance.

### ACKNOWLEDGMENTS

This work was supported by the National Natural Science Foundation of China under Grants U21A20514, 62372388 and 62071404, and by the FuXiaQuan National Independent Innovation Demonstration Zone Collaborative Innovation Platform Project under Grant 3502ZCQXT2022008.

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
