# OpenReview forum: "GLATrack: Global and Local Awareness for Open-Vocabulary Multiple Object Tracking"
_acmmm.org/ACMMM/2024/Conference — MM2024 Poster_

### Official Review · Reviewer_o6FW · 2024-05-23

**Rating:** 4
**Confidence:** 3

**Summary:**

This paper proposes a new open-vocabulary MOT method, GLATrack, which detects and tracks arbitrary classes of objects in the real world from a global and instance-level perspective. GLATrack proposes the RFE module to solve the problem of aligning image regions with region-level descriptions when using CLIP. Then, ARM is proposed for generating adaptive appearance representations for objects with different semantic richness. BSC is proposed to utilize valuable information in reference frames to resolve classification bias caused by the presence of occluding objects in key frames.

**Strengths:**

(1) For objects with different semantic richness, ARM novelly and adaptively generates different appearance representations, effectively improving the model's correlation capabilities.
(2) BSC alleviates the problem of misclassification caused by occlusion by fusing information in reference frames.
(3) GLATrack has achieved good results, and the ablation experiments of related modules are relatively sufficient, which proves the effectiveness of the method used in this article.

**Limitations:**

(1) It is known from the experimental part that among the three proposed modules, RFE plays a major role. But it seems to work similarly to the method in RegionCLIP. The author needs to explain the difference between the two.
(2) As mentioned in line 405, contrastive learning-based methods for association are deployed in RFE. The author should perform ablation experiments on the role played by this part.

**Suitability:**

3

---

### Official Review · Reviewer_KrCU · 2024-05-24

**Rating:** 3
**Confidence:** 3

**Summary:**

This paper introduces GLATrack, a novel Global and Local Awareness open-vocabulary Multi-Object Tracking (MOT) method. GLATrack integrates fine-grained semantic representations of targets from both global and instance-level perspectives. Specifically, the method includes a region-aware feature enhancement module that refines global knowledge to complement local target information, thereby bridging the distribution gap between image feature maps and pooled regional features. Additionally, this paper proposes a bidirectional semantic complementarity strategy to address semantic misalignments caused by missing target information in key frames, by dynamically selecting valuable information from reference frames to enrich object representation during the knowledge distillation process. An appearance richness measurement module is also introduced, providing appropriate representations for targets with varying appearances.

**Strengths:**

1.GLATrack introduces innovative components like the Region-aware Feature Enhancement (RFE) Module and Appearance Richness Measurement (ARM) Module that advance open-vocabulary MOT by integrating global and local semantic information to improve the tracking accuracy of diverse object types in complex scenes.
2.By leveraging transformers to refine region features and integrating bidirectional semantic complementarity, GLATrack effectively bridges the theoretical aspects of semantic understanding with practical multi-object tracking challenges. This enhances the system's ability to handle semantic misalignments and appearance variations among objects.
3.The method might inherit biases from pre-trained models used in the system, particularly in scenarios not well represented in training data. This could affect the system's performance in diverse or less common tracking scenarios.

**Limitations:**

1.In the paper, Figure 1 does not effectively illustrate the motivation behind the study, which is crucial for understanding the context and significance of the research. Additionally, the abstract could provide a clearer description of the motivation. Enhancing these aspects would not only help clarify the initial impetus for the study but also enable readers to grasp the importance and potential impact of the research more intuitively. It is recommended that the authors revise Figure 1 to better reflect the foundational concepts and motivations of the study, and refine the abstract to explicitly highlight the key drivers and objectives of the proposed method.
2.The performance enhancements reported are heavily reliant on the availability of high-quality annotations and extensive pre-training on large datasets, which might not be feasible in all real-world applications where such detailed annotations are unavailable.
3.The description of the Region-aware Feature Enhancement module in Section 2.3 is somewhat vague. Providing specific examples of how this module interacts with other components of the system or including pseudocode could enhance understanding.

**Suitability:**

3

---

### Official Review · Reviewer_ty1h · 2024-05-24

**Rating:** 4
**Confidence:** 3

**Summary:**

The paper proposes a method called GLATrack (Global and Local Awareness open-vocabulary Multi-Object Tracking) to address challenges in multi-object tracking (MOT) where arbitrary objects may be encountered beyond the training set. Traditional methods primarily focus on instance-level association and identification of novel objects, often neglecting the fine-grained semantic representations of targets within key and reference frames. GLATrack incorporates both global and instance-level perspectives to improve tracking performance.

**Strengths:**

This paper presents several strengths:
This paper introduces a novel approach to open-vocabulary multi-object tracking (MOT) by integrating global and local awareness, semantic complementarity, and appearance richness measurement. These components address the limitations of existing methods that primarily focus on instance-level association and identification of novel objects. The method achieves significant improvements in tracking accuracy compared to state-of-the-art methods, demonstrating its effectiveness in open-world scenarios where tracking arbitrary objects is essential. Specifically, it shows a 6.9% increase in TETA and a 5.6% increase in mean Average Precision (mAP) on the large-scale TAO benchmark.

**Limitations:**

While the paper addresses an important challenge in open-vocabulary multi-object tracking and introduces novel approaches, I have raised concerns regarding several aspects of the work:

1. The metrics for base+RFE in Table 4 are not significantly different from those for base+RFE+ARM, making it difficult to assess ARM's advantages. The authors should conduct additional experiments to provide further evidence.

2. In Figure 5, the authors did not explain in the text that OVTrack focuses more on objects compared to GLATrack, contrary to the description in the caption.

3. The authors need to review some spelling errors, such as "$\mathcal{L}CE$" -> "$\mathcal{L}_{CE}$" on line 528.

**Suitability:**

2

---

### Official Review · Reviewer_FKr7 · 2024-05-25

**Rating:** 3
**Confidence:** 3

**Summary:**

This paper proposed a Global and Local Awareness open-vocabulary MOT method (GLATrack), which learns to tackle the task of real-world MOT from both global and instance-level perspectives. Experiments on TAO shows its effectiveness.

**Strengths:**

1. This paper is well-organized, and the idea is reasonable, which is a incremental work indeed.
2. We propose a flexible RFE module to leverage global information and local targets, and ARM module to generate adaptive appearance representation,
3. Experiments are sufficient to validate its effectiveness.

**Limitations:**

1. This paper just improves the OVTrack from global and local awareness features, but OVTrack combines OV Object Detection and MOT.  GLATrack seems just a improvement of MOT, which lacks of novelty.
2. The RFE, BSC and ARM module are designed for the association of MOT, so are they benefical for MOT methods, such as ByteTrack, OCSORT ....

**Suitability:**

2

---

### Meta-Review · Area_Chair_v2t1 · 2024-07-01

**Recommendation:** Accept (Poster)
**Confidence:** 5

**Metareview:**

The paper was reviewed by four experts in the field and received three borderline accept and one weak reject in the final ratings. The paper focuses on the challenging open-vocabulary multi-object tracking (MOT) by integrating global and local awareness, semantic complementarity, and appearance richness measurement. Its pros including reasonable idea, good results and sufficient ablation study, significant improvements than previous sota methods like TETA, etc. Reviewer KrCU questioned the unclear motivation, reliance on high-quality annotations, and vague description of Region-aware Feature Enhancement module in Section 2.3. The AC checked the rebuttal and was convinced the authors well addressed these concerns. Given the recommendations from three reviewers, the AC recommend this paper for acceptance.

The authors are required to include the comparison with OVTrack, ablation studies, motivation, and so on in the rebuttal to the final version. A through proofreading/polishing is suggested to improve clarity and avoid typos.